# Implication of Sodium Hypochlorite as a Sanitizer in Ready-to-Eat Salad Processing and Advantages of the Use of Alternative Rapid Bacterial Detection Methods

**DOI:** 10.3390/foods12163021

**Published:** 2023-08-11

**Authors:** Alyexandra Arienzo, Valentina Gallo, Federica Tomassetti, Giovanni Antonini

**Affiliations:** 1National Institute of Biostructures and Biosystems (INBB), Viale delle Medaglie d’Oro 305, 00136 Rome, Italy; alyexandraarienzo@gmail.com; 2Department of Science, Roma Tre University, Viale Guglielmo Marconi 446, 00146 Rome, Italy; valentina.gallo3@uniroma3.it (V.G.); federica.tomassetti@uniroma3.it (F.T.)

**Keywords:** ready-to-eat vegetables, sodium hypochlorite, food sanitizers, shelf-life, microbiological quality, food safety, rapid microbiological methods

## Abstract

The use of disinfection agents in the washing processing of ready-to-eat (RTE) vegetables, especially sodium hypochlorite, is a common industrial practice performed to enhance microbiological quality. However, some studies have reported a restart of bacterial growth and a substantial increase in bacterial load during early storage associated with the use of disinfection agents, which might represent a risk for consumers. We evaluated the effect of sodium hypochlorite on bacterial growth trends during the shelf-life in *Lactuca sativa*, simulating the industrial procedures for RTE vegetable packaging. Immediately after sodium hypochlorite treatment, an effective abatement of the bacterial load was observed, followed by a restart of growth throughout storage. After 5 days, the bacterial load was close to that reached by the control samples, indicating that the net increase in bacterial load was significantly higher in the treated samples. This might be ascribed to the reduction in competitive microflora and/or to the induction of adaptive responses by resting bacteria, which might select disinfectant-resistant bacteria. These findings elicit some concerns about the actual duration of the shelf-life; products might decrease their microbiological quality earlier during storage, pointing out the need to better clarify the impact of sodium hypochlorite as a sanitizer to closer consider its use in RTE vegetable processing. Furthermore, due to the importance of the rapid estimation of bacterial load and the early detection of foodborne pathogens throughout the food chain, the accuracy of the rapid bacteria detection method, the Micro Biological Survey (MBS), and its effectiveness for microbiological analyses of RTE vegetables were evaluated.

## 1. Introduction

Ready-to-eat (RTE) products are minimally processed foods that are intended for consumption without prior preparation or cooking. They represent a great advantage in terms of technological innovation and customer satisfaction and are chosen due to the general perception of them being genuine, health-promoting and safe and because of their convenience in terms of time saving and ease of use. For these reasons, during the last years, there has been a significant increase in the market of diverse types of RTE products, including vegetables [1].

However, the processing to which RTE vegetables are subjected, although minimal, can affect the quality of the product compared to unprocessed products [2]. Indeed, toward the end of their shelf-life, the organoleptic characteristics of RTE vegetables start to decline, leading to wilting, browning and the development of off-odors and off-flavors [3]. Furthermore, microbiological quality and safety are also impaired, and both spoilage and pathogenic microorganisms have been frequently reported when the expiration date approaches [4]. Altogether, alterations occurring during the shelf-life, also including those caused by poor infrastructure in the supply chain, hinder the overall quality of RTE vegetables, reducing the product’s acceptance from consumers and leading to food waste and serious economic loss. Indeed, together with fresh vegetables, RTE vegetables account for approximately 47% of the total food waste generated by households in the European Union [5]; it has been estimated that, in the United Kingdom, 37,000 tons (178 million bags) of packaged salad are thrown away each year [6]. Due to this, great effort to find innovative treatments aimed at improving methodologies that increase RTE vegetables’ shelf-lives has been made and continues to be ongoing. In this context, investigating the alterations occurring during storage can be of great importance to find solutions to provide consumers with higher quality products and to achieve a strong reduction in unsold or wasted stock.

Industrial washing treatments, with or without the use of sanitizing agents, are one of the few measures available to maintain unaltered organoleptic features and to control the microbiological risk associated with RTE vegetables, improving quality and safety during their shelf-life [7].

A wide variety of washing treatments and chemical sanitizers have been tested with various degrees of effectiveness in reducing pathogens and spoilage microorganisms on RTE vegetables. Sodium hypochlorite is one of the most used and effective sanitizing agents for minimally processed vegetables [8] that has been proven to significantly lower the bacterial load and to be effective against various pathogens, including *L. monocytogenes* and *Salmonella* spp. [9]. Sodium hypochlorite has a low cost, is readily available and does not significantly impact the nutritional and sensorial quality of fresh products [10]. Although a high concentration of sodium hypochlorite could be harmful due to the formation of disinfection by-products (DPBs) derived from the generation of chlorine [11], studies have demonstrated that, in washing processes, effective disinfection is obtained by reaching free chlorine concentrations in the range of 50 to 200 mg/L, which are far below the allowed DBP concentrations in drinking water [12].

The effectiveness of sodium hypochlorite in reducing microbial growth in RTE vegetables, including fresh-cut salads (FCSs), is well documented. Lee et al. (2009) studied the effects of diverse concentrations of sodium hypochlorite and washing conditions on the reduction in microbiological contamination in FCS samples artificially contaminated by diverse bacterial species, including *E. coli*, *S. aureus* and total aerobic bacteria. They observed a reduction of approximately 2.0 log CFU/g in samples treated with concentrations of sodium hypochlorite in a range between 50 and 200 ppm [13]. Pan and Nakano (2014) obtained similar results, observing a reduction of 2.38 CFU/g for total plate counts and of 2.05 CFU/g for total coliform counts after the treatment of FCS with sodium hypochlorite concentrations between 50 and 200 ppm. They also observed a reduction of 2.38 CFU/g in lettuce samples inoculated with the *E. coli* O157:H7 strain in the same range of sodium hypochlorite concentrations [14]. Garcia et al. (2003) showed an aerobic plate count reduction of up to 1.4 log CFU/g of fresh lettuce samples treated with chlorine at concentrations between 100 and 200 mg/L [15].

These results, together with the reduction in water consumption during the washing treatments allowed by the usage of sodium hypochlorite, have stimulated its extensive use in the industrial processing of RTE vegetables [16,17,18].

Despite the undisputed antimicrobial effectiveness of sodium hypochlorite, studies of the effects of sodium hypochlorite throughout storage have reported controversial results. Bachelli et al. (2019) found that, compared to other sanitation agents, sodium hypochlorite was the most effective in reducing total microbial loads in fresh-cut lettuce up to the sixth day of storage [19]. However, data from other studies demonstrate that the antibacterial effects of sodium hypochlorite do not cover the entire shelf-life of RTE vegetables but are restricted only to the first days after treatment; moreover, augmented bacterial growth in the later shelf-life has been observed [20,21,22,23]. Koseki and Isobe, evaluating the changes in bacteria populations during storage at 10 °C for lettuce samples treated with sodium hypochlorite at a concentration of 200 ppm, observed that the total aerobic bacteria count was initially reduced but then increased rapidly, compared with non-treated lettuce samples [24]. Osaili et al. (2018), examining the concentration of *Enterobacteriaceae* on iceberg lettuce during post treatment storage, found that, although a reduction in bacterial counts is observed immediately after treatment with sodium hypochlorite at a concentration of 200 ppm, after the fourth day of storage, the bacterial growth on lettuce treated with sodium hypochlorite was more rapid than that on samples treated with water only, increasing by 1.5 and 2.2 log CFU/g, respectively [25]. Similar results were also obtained for yeast and molds by Li et al. (2001). They studied the effect of treatment with chlorinated water on lettuce samples in reducing yeast and mold populations. When treating iceberg lettuce with a chlorine solution at a concentration of 20 mg/L, after an initial reduction in yeast and mold loads, the restarting of bacterial growth within the second day of storage at 5 and 15 °C was observed [26].

The above discussed results open some health-impacting issues, eliciting the possibility that the use of sodium hypochlorite could instead affect the microbiological quality of RTE vegetables and constitute a risk for consumers. Indeed, the unexpected restart of bacterial growth, observed in vegetable samples treated with sodium hypochlorite already a few days after storage, suggests that the treatment with sodium hypochlorite could initially hide the presence of sub-lethally injured and still viable but not culturable bacteria (VBNC) that, under favorable conditions, might cause spoilage or regain aspects of their virulence. In this context, some researchers speculated on the possibility that treatment with sodium hypochlorite might reduce the number of competing bacteria allowing the remaining bacteria to thrive [27]. This hypothesis is supported by studies performed on endive samples treated with hydrogen peroxide. In this study, the authors observed that, after an initial post-treatment reduction in the *L. monocytogenes* load, a rapid increase occurred and demonstrated that this effect was caused by the depletion of competitive background microflora [28]. In addition, in the absence of competitive microorganisms, pathogen bacteria might emerge, and this could promote the occurrence of antimicrobial resistance, representing a threat to global health and food security.

However, despite its extensive use, studies that assess the microbiological risks associated with the use of sodium hypochlorite as a sanitizing agent for RTE vegetables are still scarce, and further investigations are needed to provide a comprehensive explanation of this issue because of its potential impact on public health. Our study was motivated by these needs, in an attempt to contribute to the improvement of knowledge in this field and to find new solutions to deal with its implications. Indeed, the aim of this work was the assessment of patterns of bacterial growth trends (i.e., total aerobic mesophilic bacteria and Enterobacteriaceae) during storage and in samples of *Lactuca sativa* treated with sodium hypochlorite, as well as the evaluation of alternative detection methods able to provide rapid identification and quantification of VBNC bacteria.

The sodium hypochlorite was used at a concentration of 0.22 g/L. The microbiological analyses were performed before the treatment and immediately after and throughout storage, simulating industrial packaging and storing samples at a home refrigeration temperature (i.e., 4 °C), using reference plate count methods. In addition, microbiological analyses were also performed by using the Micro Biological Survey (MBS) method, a colorimetric system for the rapid detection and quantification of bacteria that measures the catalytic activity of redox enzymes in the main metabolic pathways of bacteria, allowing for an unequivocal correlation between enzymatic activity and bacterial concentration [29]. The MBS was previously proven to be accurate for the rapid detection of total aerobic mesophilic bacteria in RTE vegetables [30]. Indeed, a further aim of this work was the evaluation of the MBS’s accuracy for the rapid detection of *Enterobacteriaceae* in RTE salads. This can be useful in an attempt to extend the MBS’s area of applicability to the assessment of the microbiological quality of RTE salads. This potential application of the MBS method could be advantageous, especially in contexts in which classic methods give poor results in terms of sensitivity, such as when rapid detection of VBNC bacteria is needed, and in terms of rapidity, such as in the case of industrial contexts that rely on a fast evaluation of the overall microbiological quality and/or of the efficacy of decontamination treatments throughout the entire food chain.

The results obtained in this study suggest that the use of sodium hypochlorite could affect the microbiological quality of RTE vegetables, presumably favoring the growth of disinfectant-resistant bacteria, opening potential health-impacting concerns and the need to more closely consider its use in RTE vegetable processing. In addition, we demonstrated the applicability of the MBS method to the assessment of the microbiological quality of RTE salads and its potential for the rapid detection of resting VBNC bacteria, which represent a great concern in minimally processed foods.

## 2. Materials and Methods

Raw samples. A total of 70 heads of *L. sativa* were selected among different retailers in the city of Rome, Italy. Samples were collected within 2 h from unloading and were transported under refrigeration conditions (4 ± 1 °C) to the laboratory, where they were immediately processed according to good manufacturing practices and analyzed. The effects of washing treatments in the absence and presence of sodium hypochlorite on bacterial growth were evaluated for both non-packaged fresh-cut samples immediately after processing and throughout storage, simulating industrial packaging.

Sample preparation and processing. For samples intended to be analyzed immediately after processing, 30 heads of *L. sativa* were used, and each was divided into three samples of 150 g. The samples were treated as follows. Control (CTRL): Samples were immediately analyzed without further treatment. Treatment 1: Samples were washed thoroughly in water for about 2 min while gently scrubbing the surface of the leaves. Treatment 2: Samples were immersed in a 0.22 g/L aqueous solution of sodium hypochlorite. After 15 min of immersion, the samples were washed thoroughly in water for about 2 min while gently scrubbing the surface of the leaves. Prior analyses of all samples, including the control, were prepared by homogenizing 30 g in 275 mL of Buffered Peptone Water (BPW, AppliChem, Darmsdadt, Germany) using a Stomacher 400 (Seward, London, UK) for 120 s at medium speed with serial dilution in the same diluent when needed.

Samples that were intended to be analyzed throughout storage were prepared and processed as follows. A total of 40 heads of *L. sativa* were divided into two samples of 300 g each. The first sample (300 g), used as the control, was washed thoroughly in water for about 2 min while gently scrubbing the surface of the leaves, spin dried and then divided into three aliquots of 100 g each, in sterile conditions. One aliquot was immediately analyzed, and the other two were packaged in sterile conditions in sterile bags for food handling, stored at a refrigerated temperature and analyzed after 2 and 5 days. The second sample (300 g) was treated with sodium hypochlorite as previously described, spin dried and divided into three aliquots (100 g each) in sterile conditions. The first sample was immediately analyzed, and the other two were packaged in sterile conditions in sterile bags for food handling, stored at a refrigerated temperature and analyzed after 2 and 5 days.

Microbiological analysis. Total aerobic mesophilic bacteria and *Enterobacteriaceae* were chosen as general microbiological quality indicators. Analyses were performed using the reference plate count methods, according to UNI EN ISO 4833-1:2013 for the total aerobic mesophilic count (TAMC) and ISO 21528:2017 for *Enterobacteriaceae*.

In parallel, analyses were also performed using the MBS method. The MBS method is a colorimetric system used for the detection and quantification of bacteria in food and water samples. Analyses using the MBS method were performed using MBS Total Viable Count (TVC) and EB (*Enterobacteriaceae*) vials, containing the specific lyophilized growth medium for the detection and quantification of bacterial targets.

To start the analysis, vials were rehydrated with 10 mL of sterile distilled water and paraffin oil and shaken until all of the reagent was dissolved. Vials were inoculated with 1 mL of sample homogenate and its serial dilutions, in parallel with the reference pour plate method. All analyses were performed in triplicate. Vials were incubated at 30 °C for 30 h for TAMC and 37 °C for 30 h for *Enterobacteriaceae*.

The vials’ medium color was periodically controlled with a thermostatic colorimeter that automatically detects color changes. A color change from blue to yellow in the reaction medium is indicative of a positive result, i.e., the presence of aerobic mesophilic bacteria [29]. The time of the color change after the inoculum varies according to the bacterial concentration. The time of the color change was inversely related to the bacterial content of the analyzed sample; as the bacterial concentration increased, the time required for color change decreased. The persistence of the starting color indicates a negative result, i.e., the absence of the microorganisms of interest.

For TAMC, the correlation parameters used in this study were those obtained in the work by Arienzo et al. (slope = −0.30; intercept = 8.86) [30].

For *Enterobacteriaceae*, linearity was evaluated according to ISO 16140:2016. Linearity parameters (slope and intercept) were used to evaluate the accuracy of the method.

Accuracy of the MBS method. Accuracy is the degree of correspondence between the response obtained with the reference method and the response obtained with the alternative method on identical samples (ISO 16140:2016). To demonstrate the accuracy of the MBS method, the results obtained with the MBS method (log CFU/mL) were compared to those obtained with the reference method (log CFU/mL). Regression parameters were calculated according to ISO 16140:2016. Prediction intervals were calculated using Microsoft Excel 2007 for Windows XP.

Statistical analysis. Data were analyzed statistically with a paired-samples *t*-test. A significant difference was considered at *p* < 0.05.

## 3. Results

### 3.1. Evaluation of Bacterial Growth after SAMPLES’ TREATMENT with Sodium Hypochlorite before Storage

In the first part of the experimentation, the evaluation of the antibacterial effects of sodium hypochlorite on samples of *L. sativa* before storage was performed. To this purpose, counts of viable aerobic bacteria (TAMC) and *Enterobacteriaceae* were performed, using reference plate count methods, on fresh-cut samples that were immersed for 15 min in a 0.22 g/L aqueous solution of sodium hypochlorite and then rinsed under flowing water. The obtained results were compared to results from unprocessed samples, which were used as the control, and to those obtained after simple rinsing of the samples under flowing water. The results are displayed in Figure 1.

The average bacterial concentration in the control samples (unprocessed) was 6.86 ± 0.49 and 4.08 ± 0.41 log CFU/g, respectively, for TAMC and *Enterobacteriaceae*. Washing under flowing water reduced the concentration of TAMC to 6.2 log CFU/g with a bacterial load reduction of 0.7 log units, but it did not impact the concentration of *Enterobacteriaceae*.

As expected, the treatment with sodium hypochlorite was more effective in reducing both TAMC and *Enterobacteriaceae* compared to simple rinsing, although variability was observed among the samples. Indeed, after treatment with sodium hypochlorite, the bacterial concentration decreased to 5.1 log CFU/g for total aerobic mesophilic bacteria and to 2.9 log CFU/g for *Enterobacteriaceae*, with a bacterial load reduction of 1.9 log units and 1.2 log units, respectively, compared to the control.

### 3.2. Evaluation of Bacterial Growth after Samples’ Treatment with Sodium Hypochlorite throughout Storage

To evaluate the impact of the sodium hypochlorite treatment on bacterial growth trends throughout storage, samples of *L. sativa* were treated with sodium hypochlorite (as previously described), packaged under sterile conditions in sterile bags for food handling and stored at a refrigerated temperature (4 °C), as described in detail in Section 2. In this case, samples that were simply rinsed under flowing water (without sodium hypochlorite) were used as controls. All samples, including the controls, were analyzed immediately before packaging and after 2 and 5 days of storage. The results are shown in Figure 2.

For TAMC, the results from the samples analyzed immediately before packaging (t = 0) displayed a bacterial load of 5.9 log CFU/g for the control samples and of 4.6 log CFU/g for sodium-hypochlorite-treated samples, showing a bacterial load reduction of 1.3 log units of the treated samples compared to the control (Figure 2a). Similar results were observed for the *Enterobacteriacee* count, showing the same rate of bacterial load reduction in treated samples compared to the control (1.3 log units) (Figure 2b). These data confirm the data from the previous experiment (Figure 1), showing the effectiveness of sodium hypochlorite in reducing bacterial load. However, data from samples analyzed throughout storage showed different results. Indeed, the rapid restart of bacterial growth, which was similar for TAMC and *Enteriobacteriacee*, was observed for control samples but also for sodium-hypochlorite-treated samples at t = 2 and t = 5.

For TAMC, at t = 2, the bacterial load of the control and treated samples increased, compared to t = 0, reaching a concentration of 6.3 and 5.6 log CFU/g, respectively. After 5 days of storage (t = 5), a further increase in the total aerobic mesophilic bacteria load was observed for both the control and treated samples, showing concentration values of 7.1 and 7.0 log CFU/g, respectively. It is noteworthy that the total increase in bacterial load (from t = 0 to t = 5) was higher for treated samples (2.4 log units) than that for control samples (1.2 log units) (Figure 2a).

For *Enterobacteriaceae*, at t = 2, the bacterial concentration increased from 4.8 (t = 0) to 5 log CFU/mL for the control and from 3.5 (t = 0) to 4.1 log CFU/g for treated samples, reaching after 5 days of storage (t = 5) the concentrations of 5.6 and 5.2 log CFU/g for the control and treated samples, respectively. Even in this case, the total increase in bacterial load was higher for treated samples compared to the control, being 1.7 and 0.8 log units, respectively.

### 3.3. Accuracy of the MBS Method

In this study, the TAMC and Enterobacteriaceae analyses on *L. sativa* samples were performed with the reference method and the alternative MBS method, in all experimental conditions.

To evaluate the linearity of the method, i.e., the ability of the method when used with a given matrix to give results that are in proportion to the amount of analyte present in the sample, the bacterial concentrations (expressed as the log of CFU/mL) obtained with the reference method were compared with the time at which a color change occured in identical L. sativa samples analyzed with the MBS method.

For TAMC, the linearity of the MBS method was previously demonstrated for RTE salad samples and was further confirmed in this study (slope = 0.30; maximum analysis time = 30 h; R^2^ = 0.81) [30]. The maximum time needed to obtain results with the MBS method was 20 h, and 90% of the results were obtained between 5 and 14 h.

For *Enterobacteriaceae*, the linearity of the MBS method was evaluated according to ISO 16140:2016. A linear inverse relationship between the time required for a color change in the MBS EB vials and *Enterobacteriaceae* concentration (log CFU/mL) was observed (slope = 0.27; maximum analysis time = 24 h; R^2^ = 0.82). The maximum time needed to obtain results with the MBS method was 20 h, and 80% of results were obtained between 8 and 15 h.

The regression parameters were used to evaluate the accuracy of the MBS method for the detection and quantification of TAMC and *Enterobacteriaceae*.

Accuracy was evaluated according to ISO 16140:2016. Figure 3a,b show the linear regression analyses for TAMC (slope = 1.0096; R^2^ = 0.81) and *Enterobacteriaceae* (slope = 1.00; R^2^ = 0.82), respectively.

## 4. Discussion

The use of disinfection agents in the washing and processing of RTE vegetables is a common industrial practice used to improve the microbiological quality of these foods. One of the most used disinfection agents is sodium hypochlorite because of its proven antimicrobial effectiveness, low cost and negligible impact on the organoleptic properties of fresh-cut vegetables [16,17,18].

Even if many studies have demonstrated the effectiveness of sodium hypochlorite in reducing microbial growth in RTE vegetables, interesting results have come from other, although fewer, studies that conversely showed the restart of bacterial growth and a higher increase in the bacterial load of treated samples compared to controls after a few days of storage. However, the results are still controversial due to the current scarcity of studies on this topic.

In this work, we simulated the industrial packaging of fresh-cut samples of *L. sativa* and home storage conditions with the aim of evaluating the effect of sodium hypochlorite on bacterial growth trends during their shelf-life, using *Enterobacteriaceae* and total aerobic mesophilic bacteria as microbiological quality indicators. The microbiological analyses were performed using the reference plate count methods in parallel with the Micro Biological Survey (MBS) method, which, here, we demonstrated to have a high accuracy for both TAMC and *Enterobacteriaceae* and the advantage of providing faster results compared to the traditional plate count methods, thus showing great potential in the microbiological quality control of RTE vegetables before and after treatment.

Interestingly, our data support the data from previous studies, showing similar patterns of reductions in and restarting of bacterial loads during storage. Indeed, our results show the effective abatement of bacterial loads during earlier times after treatment (t = 0), followed by the restarting of bacterial growth (already observed at t = 2) and a significant increase in bacterial load up until the end of the storage period (t = 5). Importantly, the increase in bacterial growth during storage was significantly higher in samples treated with sodium hypochlorite compared to the control samples.

Even if the causes underlying these sodium hypochlorite treatment effects are not yet known, they might be ascribed to a reduction in competitive microflora or to the induction of adaptive responses by resting bacteria; moreover, treatment with sodium hypochlorite might cause plant tissue damage, enhancing microbial proliferation. Several authors have investigated the stress responses of Gram-negative bacteria to sodium hypochlorite and its active ingredient, hypochlorous acid, detecting the occurrence of disinfectant-resistant bacteria, which have also been found in fresh-cut vegetables [31]. Although the molecular mechanisms involved in this stress response are not completely understood, it has been demonstrated that, to adapt to the damage induced by hypochlorous acid, microorganisms alter their metabolism and induce several adaptive responses, such as the augmented transcription of genes related to membrane and DNA damage repair, the expression of detoxifying enzymes and alterations in energy metabolism. Moreover, changes in the structural stability of cell membranes and in membrane permeability caused by sodium hypochlorite have been documented to induce the formation of biofilms, which is another important adaptive response [32]. The presence of sub-lethally injured and VBNC bacteria is of great interest in minimally processed foods, such as fresh products and FCS, because, despite them not being detectable with traditional methods, they still are able to pose a risk to consumers. Under favorable conditions, they are again able to gain their distinctive features and cause spoilage or regain aspects of their virulence. In this context, the use of the MBS method could be advantageous, because, being based on the measures of bacterial metabolic activity, it could show a higher sensitivity for the detection of VBNC bacteria. Moreover, disinfectant-resistant bacteria contamination appears to be an issue that should not be underestimated. The effects of chlorine treatment on bacterial survival and growth are currently being studied due to their involvement in triggering entrance into the VBNC state of important food pathogens, such as *Salmonella* spp. and *Listeria monocytogenes* [33], and due to increasing the number of resistant bacteria [34], posing a significant threat to public health.

Given the results obtained in this study, we believe that the effects of sodium hypochlorite on the survival and growth kinetics of bacteria contaminating fresh-cut lettuce should be more closely considered for the safer use of this sanitizing agent in the processing and production of RTE vegetables.

## 5. Conclusions

The results obtained in this study show that treatment with sodium hypochlorite could affect the microbiological quality of RTE salads. This could lead to a potential health-impacting scenario in which sodium hypochlorite, instead of guaranteeing and improving microbiological quality, might be a bacterial growth promoter during products’ shelf-lives.

Based on this, we believe that, to ensure the microbiological and sensory quality of fresh-cut lettuce and increase shelf-life, further investigations are needed that better clarify the impact of sodium hypochlorite as a sanitizer in RTE vegetable processing and production, and that allow for the better evaluation of its usage. In this context, we highlight the need to more deeply investigate the stress responses of contaminating bacteria after sanitizing treatments, and we point out the importance of seeking alternative decontamination processes that are able to act also during storage. Indeed, several emerging disinfection strategies, including non-thermal physical treatments, such as UV light, ionizing radiation, high hydrostatic pressure and high-intensity ultrasound, especially when used in combination, have been demonstrated to be efficacious in ensuring the high microbiological quality of RTE vegetables [35,36,37]. Furthermore, the use of these treatments can also be useful to minimize the risk of product deterioration, such as in the case of ultrasonic treatments that have been demonstrated to inhibit the activity of PPO, POD and cell-wall-degrading enzymes, thus preventing vegetable browning [38].

In addition, because the research is currently focusing on the development of timely high-throughput microbiological screening methods to detect critically contaminated samples [39], in this study, we also investigated the possible application of the rapid bacterial detection method MBS on RTE vegetables, demonstrating its accuracy for both total aerobic mesophilic bacteria and *Enterobacteriaceae* detection and the potential advantages of its use in industrial applications. Indeed, reducing analytical times and eliminating the need for external laboratories can be of great relevance for microbiological monitoring in the food business context, including in the industrial production of RTE foods. Furthermore, in contrast with traditional methods that rely on colony plate counts, the MBS method measures the catalytic activity of the redox enzymes of the bacterial respiratory chain, and this could be exploited also for the detection of VBNC bacteria that constitute a potential concern in minimally processed foods.

## Figures and Tables

**Figure 1 foods-12-03021-f001:**
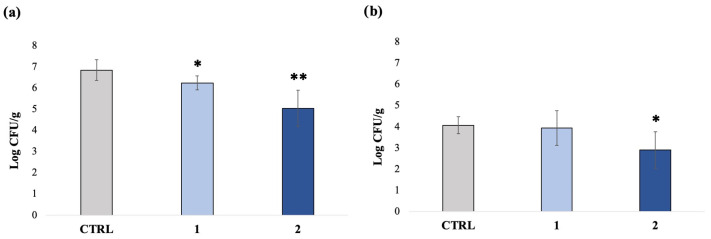
*L. sativa* average contamination levels under different experimental conditions evaluated using the plate count method: (**a**) TAMC and (**b**) *Enterobacteriaceae*. CTRL: Unprocessed; 1: samples rinsed under flowing water in absence of sodium hypochlorite (treatment 1); 2: samples immersed in water bath with sodium hypochlorite (0.22 g/L) followed by rinsing under flowing water (treatment 2) (SD < 10%). * Significant difference compared to “CTRL” (*p* < 0.05); ** Significant difference compared to “CTRL” and compared to “treatment 1” (*p* < 0.05); n = 30.

**Figure 2 foods-12-03021-f002:**
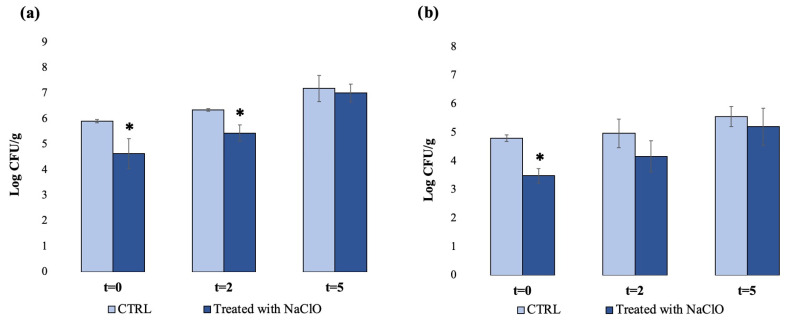
Average contamination levels of *L. sativa* samples at different storage times, evaluated using the plate count method: (**a**) TAMC and (**b**) *Enterobacteriaceae* (SD < 10%). * Significant difference (*p* < 0.05); n = 40. Control samples were rinsed under flowing water, and the treated samples were immersed in a 0.22 g/L aqueous solution of sodium hypochlorite (NaClO) and then rinsed under flowing water. The analyses were performed immediately before packaging (t = 0) and after 2 and 5 days (t = 2; t = 5) of storage.

**Figure 3 foods-12-03021-f003:**
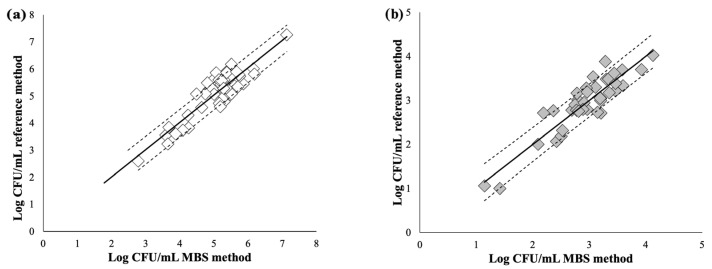
Correlation lines between the results obtained with the MBS method and reference methods for TAMC (**a**) and *Enterobacteriaceae* (**b**). The quantitative results obtained with the reference method were plotted against the results obtained with the MBS method. Regression parameters of TAMC (slope = 1.00, R^2^ = 0.8, n = 50) *Enterobacteriaceae* (slope = 1.00, R^2^ = 0.82, n = 50). Dashed lines represent 90% prediction intervals.

## Data Availability

The data used to support the findings of this study can be made available by the corresponding author upon request.

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
