# Peer review of "Implication of Sodium Hypochlorite as a Sanitizer in Ready-to-Eat Salad Processing and Advantages of the Use of Alternative Rapid Bacterial Detection Methods"

_foods, 2023, doi:10.3390/foods12163021_

Round 1

Reviewer 1 Report

Reviewer comments: 02.08.2023

A manuscript with the title: "Implication of sodium hypochlorite as a sanitizing agent in the processing of ready-to-eat salads and the advantages of using alternative methods for rapid detection of bacteria" by Alyexander Arienzo, Valentin Gallo, Federic Tomassetti and Giovanni Antonini, is an interesting look at the sterilization of ready to eat (RTE) vegetables and their shelf – life by health of peoples. The work is important in terms of safety and uses modern bacterial identification techniques that are efficient and effective to perform in relation to previously existing ones. Reducing analytical times and eliminating the need of external laboratories can be of great relevance for microbiological monitoring in the food business context, including the industrial production of RTE foods. Furthermore, in contrast with traditional methods that rely on colony plate count, the MBS method measures the catalytic activity of redox enzymes of bacterial respiratory chain, and this could be exploited also for the detection of VBNC bacteria that constitute a potential concern in minimally processed foods.

Decision: Minor revision

General conclusions, suggestions for the authors:

1.     The manuscript is written in correct English, there are a few typos probably due to autocorrect.

2.     Abstract and Introduction, Results, Discussion and Material and Methods are written correctly except for small errors that the authors will be able to correct very quickly, which are listed below:

Line 20 – “ob – served  exchange to “observed”. It's not a mistake, but it looks better in text and is easier to read.

Line 29 – “food – borne  exchange to “foodborne”.

Line 45 – “to – wards” exchange to “towards”.

Line 46 – “de – cline” exchange to “decline”.

Line 52 – “veg – etables” exchange to “vegetables”.

Line 57 – “investi – gating” exchange to “investigating”.

Line 72 – “dis infection” exchange to “disinfection”.

Line 77 – “veg – etables” exchange to “vegetables”.

Line 79 – “reduc­ – tion” exchange to “reduc­tion”.

Line 82 – “so – dium” exchange to “sodium”.

Line 85 – “concentra – tions” exchange to “concentrations”.

Line 89 – “ef – fects” exchange to “effects”.

Line 102 – “hy – pochlorite” exchange to “hypochlorite”.

Line 113 – “restart – ing” exchange to “restarting”.

Line 118 – “al ready” exchange to “already”.

Line 120 – “cultura – ble” exchange to “culturable”.

Line 121 – “as – pects” exchange to “aspects”.

Line 125 – “ob – served  exchange to “observed”.

Line 126 – “in crease” exchange to “increase”.

Line 127 – “compet – itive” exchange to “competitive”.

Line 128 – “microorgan – isms” exchange to “microorganisms”.

Line 129 – “antimi crobial” exchange to “antimicrobial”.

Line 137 – “home – refrigeration” exchange to “home refrigeration”.

Line 139 – “Bio – logical” exchange to “Biological”.

Line  140 – “quantifi – cation” exchange to “quantification”.

Line 141 – “meta – bolic” exchange to “metabolic”.

Line 142 – “activ – ity” exchange to “activity”.

Line 144 – “fur – ther” exchange to “further”.

Line 148 – “clas – sic” exchange to “classic”.

Line 151 – “decon – tamination” exchange to “decontamination”.

Line 157 – “trans – ported” exchange to “transported”.

Line 158 – “immedi ately” exchange to “immediately”.

Line 167 – “wa – ter” exchange to “water”.

Line 189 – “ref – erence” exchange to “reference”.

Line 190 – “meso – philic” exchange to “mesophilic”.

Line 195 – “me dium” exchange to “medium”.

Line 208 – “ab – sence” exchange to “absence”.

Line 210 – “Ar – ienzo” exchange to “Arienzo”.

Line 212 – “Lin – earity” exchange to “Linearity”.

Line 216 – “alterna – tive” exchange to “alternative”.

Line 223 – “Signifi cant” exchange to “Significant”.

Line 230 – “us – ing” exchange to “using”.

Line 248 – “reduc – ing” exchange to “reducing”.

Line 250 – “hypo – chlorite” exchange to “hypochlorite”.

Line 257 – “previ ously” exchange to “previously”.

Line 260 – “so – dium” exchange to “sodium”.

Line 261 – “ana – lyzed” exchange to “analyzed”.

Line 272 – “dis – played” exchange to “displayed”.

Line 273 – “so – dium” exchange to “sodium”.

Line 275 – “Enter obacteriacee” exchange to “Enterobacteriacae”.

Line 278 – “re ducing” exchange to “reducing”.

Line 281 – “Enteriobacteriacee”  exchange to “Enterobacteriacae”.

Line 283 – “com pared” exchange to “compared”.

Line 285 – “ob – served  exchange to “observed”.

Line 297 – “condi tions” exchange to “conditions”.

Line 302 – “iden – tical” exchange to “identical”.

Line 317 – “Enteriobac – teriacae”  exchange to “Enterobacteriacae”.

Line 320 – “meth – ods” exchange to “methods”.

Line 323 – “rep – resent” exchange to “represent”.

Line 329 – “veg – etables” exchange to “vegetables”.

Line 331 – “re – ducing” exchange to “reducing”.

Line 334 – “stor age” exchange to “storage”.

Line 339 – “meso – philic” exchange to “mesophilic”.

Line 343 – “tra – ditional” exchange to “traditional”.

Line 350 – “in – crease” exchange to “increase”.

Line 355 – “hy – pochlorite” exchange to “hypochlorite”.

Line 357 – “hypo – chlorite” exchange to “hypochlorite”.

Line 358 – “disinfect ant” exchange to “disinfectant”.

Line 360 – “under stood” exchange to “understood”.

Line 363 – “ex – pression” exchange to “expression”.

Line 372 – “bac terial” exchange to “bacterial”.

Line 373– “bac teria” exchange to “bac teria”.

Line 379 – “hypo – chlorite” exchange to “hypochlorite”.

Line 381 – “produc – tion” exchange to “production”.

Line 385 – “hypo – chlorite” exchange to “hypochlorite”.

Line 388 – “dur – ing” exchange to “during”.

Line 392 – “produc – tion” exchange to “production”.

Line 393 – “im – portance” exchange to “importance”.

Line 398 – “sam – ples” exchange to “samples”.

Line 400 – “aero – bic” exchange to “aerobic”.

Line 407 – “con – cern” exchange to “concern”.

Line 412 – “Investiga – tion” exchange to “Investiga – tion”

Line 415 – “De – partment” exchange to “Department”.

Line 418 – “Com – ponente” exchange to “Componente”.

Author Response

Point 1. The manuscript is written in correct English, there are a few typos probably due to autocorrect.

Response 1. We thank the Reviewer for this comment aimed at improving the quality of the manuscript. We corrected the text according to the Reviewer’s suggestions.

Point 2. Abstract and Introduction, Results, Discussion and Material and Methods are written correctly except for small errors that the authors will be able to correct very quickly, which are listed below:

Line 20 – “ob – served” exchange to “observed”. It's not a mistake, but it looks better in text and is easier to read.

Line 29 – “food – borne” exchange to “foodborne”.

Line 45 – “to – wards” exchange to “towards”.

Line 46 – “de – cline” exchange to “decline”.

Line 52 – “veg – etables” exchange to “vegetables”.

Line 57 – “investi – gating” exchange to “investigating”.

Line 72 – “dis – infection” exchange to “disinfection”.

Line 77 – “veg – etables” exchange to “vegetables”.

Line 79 – “reduc­ – tion” exchange to “reduc­tion”.

Line 82 – “so – dium” exchange to “sodium”.

Line 85 – “concentra – tions” exchange to “concentrations”.

Line 89 – “ef – fects” exchange to “effects”.

Line 102 – “hy – pochlorite” exchange to “hypochlorite”.

Line 113 – “restart – ing” exchange to “restarting”.

Line 118 – “al – ready” exchange to “already”.

Line 120 – “cultura – ble” exchange to “culturable”.

Line 121 – “as – pects” exchange to “aspects”.

Line 125 – “ob – served” exchange to “observed”.

Line 126 – “in – crease” exchange to “increase”.

Line 127 – “compet – itive” exchange to “competitive”.

Line 128 – “microorgan – isms” exchange to “microorganisms”.

Line 129 – “antimi – crobial” exchange to “antimicrobial”.

Line 137 – “home – refrigeration” exchange to “home refrigeration”.

Line 139 – “Bio – logical” exchange to “Biological”.

Line 140 – “quantifi – cation” exchange to “quantification”.

Line 141 – “meta – bolic” exchange to “metabolic”.

Line 142 – “activ – ity” exchange to “activity”.

Line 144 – “fur – ther” exchange to “further”.

Line 148 – “clas – sic” exchange to “classic”.

Line 151 – “decon – tamination” exchange to “decontamination”.

Line 157 – “trans – ported” exchange to “transported”.

Line 158 – “immedi – ately” exchange to “immediately”.

Line 167 – “wa – ter” exchange to “water”.

Line 189 – “ref – erence” exchange to “reference”.

Line 190 – “meso – philic” exchange to “mesophilic”.

Line 195 – “me – dium” exchange to “medium”.

Line 208 – “ab – sence” exchange to “absence”.

Line 210 – “Ar – ienzo” exchange to “Arienzo”.

Line 212 – “Lin – earity” exchange to “Linearity”.

Line 216 – “alterna – tive” exchange to “alternative”.

Line 223 – “Signifi – cant” exchange to “Significant”.

Line 230 – “us – ing” exchange to “using”.

Line 248 – “reduc – ing” exchange to “reducing”.

Line 250 – “hypo – chlorite” exchange to “hypochlorite”.

Line 257 – “previ – ously” exchange to “previously”.

Line 260 – “so – dium” exchange to “sodium”.

Line 261 – “ana – lyzed” exchange to “analyzed”.

Line 272 – “dis – played” exchange to “displayed”.

Line 273 – “so – dium” exchange to “sodium”

Line 275 – “Enter – obacteriacee” exchange to “Enterobacteriacae”.

Line 278 – “re – ducing” exchange to “reducing”.

Line 281 – “Enteriobacteriacee” exchange to “Enterobacteriacae”.

Line 283 – “com – pared” exchange to “compared”.

Line 285 – “ob – served” exchange to “observed”.

Line 297 – “condi – tions” exchange to “conditions”.

Line 302 – “iden – tical” exchange to “identical”.

Line 317 – “Enteriobac – teriacae” exchange to “Enterobacteriacae”

Line 320 – “meth – ods” exchange to “methods”.

Line 323 – “rep – resent” exchange to “represent”

Line 329 – “veg – etables” exchange to “vegetables”.

Line 331 – “re – ducing” exchange to “reducing”.

Line 334 – “stor – age” exchange to “storage”.

Line 339 – “meso – philic” exchange to “mesophilic”.

Line 343 – “tra – ditional” exchange to “traditional”.

Line 350 – “in – crease” exchange to “increase”

Line 355 – “hy – pochlorite” exchange to “hypochlorite”

Line 357 – “hypo – chlorite” exchange to “hypochlorite”.

Line 358 – “disinfect – ant” exchange to “disinfectant”.

Line 360 – “under – stood” exchange to “understood”.

Line 363 – “ex – pression” exchange to “expression”.

Line 372 – “bac – terial” exchange to “bacterial”.

Line 373– “bac – teria” exchange to “bac – teria”.

Line 379 – “hypo – chlorite” exchange to “hypochlorite”.

Line 381 – “produc – tion” exchange to “production”.

Line 385 – “hypo – chlorite” exchange to “hypochlorite”.

Line 388 – “dur – ing” exchange to “during”.

Line 392 – “produc – tion” exchange to “production”.

Line 393 – “im – portance” exchange to “importance”.

Line 398 – “sam – ples” exchange to “samples”.

Line 400 – “aero – bic” exchange to “aerobic”.

Line 407 – “con – cern” exchange to “concern”.

Line 412 – “Investiga – tion” exchange to “Investiga – tion”

Line 415 – “De – partment” exchange to “Department”.

Line 418 – “Com – ponente” exchange to “Componente”.

Response 2. We thank the Reviewer for this comment. According to Reviewer’s suggestion, we turned off the automatic hyphenation that was the cause of the reported errors.

Reviewer 2 Report

The article "Implication of sodium hypochlorite as sanitizer in ready to eat salads processing and advantages of the use of alternative rapid 3 bacterial detection methods" has been reviewed and the following has been observed

In the introduction it is necessary to strongly highlight the contribution of the research, since as you have commented in the same manuscript, the use of sodium hypochlorite has been widely used for decades, so it is not clear what is new in this article, which calls into question its originality.

On the other hand, minor errors have been seen in the editing of the article, they have been marked in the file attached to these comments.

In the conclusions it would be important to highlight the new pretreatment trends for food disinfection as well as perspectives of this research.

In general terms, the investigation was well conducted, particularly I believe that the findings of this investigation should be highlighted much more.

some writing errors were detected, I suggest that you thoroughly review the entire document to make sure of the good writing. Additionally, there is inconsistency in the units used, especially in liters or milliliters... ml instead of mL or g/l instead of m/L

Author Response

Point 1. In the introduction it is necessary to strongly highlight the contribution of the research, since as you have commented in the same manuscript, the use of sodium hypochlorite has been widely used for decades, so it is not clear what is new in this article, which calls into question its originality.

Response 1. We thank the Reviewer for this comment aimed at improving the quality of the manuscript. We improved the manuscript according to the Reviewer’s suggestion providing the text with the followings:

“…However, despite its extensive use, studies which assess the microbiological risks associated with the use of sodium hypochlorite as sanitizing agent for RTE vegetables are still scarce and further investigations are needed to provide a comprehensive explaining of this issue, since its potential impact on public health. Our study was motivated by these needs, in the attempt to contribute to the improvement of knowledge in this field and finding new solutions to deal with its implications. Indeed, the aim of this work was the assessment of patterns of bacterial growth trend (i.e., total aerobic mesophilic bacteria and Enterobacteriaceae), during storage, in samples of Lactuca sativa treated with sodium hypochlorite and the evaluation of alternative detection methods able to provide a rapid identification and quantification of VBNC bacteria.” (Lines 134 – 143).

“…The results obtained in this study suggest that the use of sodium hypochlorite could affect the microbiological quality of RTE vegetables, presumably favouring the growth of disinfectant resistant bacteria, opening to potential health impacting concerns and to the need of closer considering its use in RTE vegetables processing. In addition, we demonstrated the applicability of the MBS method for the microbiological quality's assessment of RTE salads and its potential for the rapid detection of resting VBNC bacteria.” (Lines 162 – 168).

Point 2. On the other hand, minor errors have been seen in the editing of the article, they have been marked in the file attached to these comments.

Response 2. We corrected the manuscript and the figure 3 according to the Reviewer’s suggestions and provided the text with the following:

“…Altogether, the alterations occurring during shelf life, also including those caused by a poor infrastructure in the supply chain, hinder the overall quality of RTE vegetables…” (Lines 50 – 51).

Point 3. In the conclusions it would be important to highlight the new pretreatment trends for food disinfection as well as perspectives of this research.

Response 3. According to the Reviewer’s suggestion, we provided the conclusion with the new pretreatment trends and perspectives and added some additional references as follows:

“…we point out the importance of seeking alternative decontamination processes able to act also during storage. Indeed, several emerging disinfection strategies, including non-thermal physical treatments, such as UV-light, ionizing radiation, high hydrostatic pressure, and high-intensity ultrasound, especially when used in combination, have been demonstrated to be efficacy in ensuring high microbiological quality of RTE vegetables (Deng et al., 2020, doi: 10.1080/10408398.2019.1649633; Pinela et al., 2017, doi: 10.1080/10408398.2015.1046547; Khade et al., 2023, doi: 10.1007/s13197-023-05754-8.). Furthermore, the use of these treatments can also be useful to minimize the risk of products’ deterioration, as in the case of ultrasonic treatments that have been demonstrated to inhibit the activity of PPO, POD and cell wall-degrading enzymes, thus preventing vegetables’ browning (Jiang et al., 2020 doi: 10.1016/j.ultsonch.2020.105261)”. (Lines 413 – 420).

Point 4. In general terms, the investigation was well conducted, particularly I believe that the findings of this investigation should be highlighted much more.

Response 4. We thank the Reviewer for this comment. We modified the manuscript accordingly.

Point 5. Comments on the Quality of English Language: some writing errors were detected, I suggest that you thoroughly review the entire document to make sure of the good writing. Additionally, there is inconsistency in the units used, especially in liters or milliliters... ml instead of mL or g/l instead of m/L.

Response 5. We modified the text according to the Reviewer’s requests.
